# Domain Private Transformers for Multi-Domain Dialog Systems

**Anmol Kabra** [*]
Toyota Technological Institute
Chicago, IL
anmol@ttic.edu

**Ethan R. Elenberg**
ASAPP Inc
New York, New York
eelenberg@asapp.com

## Abstract

Large, general purpose language models have demonstrated impressive performance across many different conversational domains. While multi-domain language models achieve low overall perplexity, their outputs are not guaranteed to stay within the domain of a given input prompt. This paper proposes *domain privacy* as a novel way to quantify how likely a conditional language model will leak across domains. We also develop policy functions based on token-level domain classification, and propose an efficient fine-tuning method to improve the trained model's domain privacy. Experiments on membership inference attacks show that our proposed method has comparable resiliency to methods adapted from recent literature on differentially private language models.

## 1 Introduction

Large language models have enabled significant progress in machine learning and NLP across a wide range of tasks and domains (Bommasani et al., 2021). They perform especially well in settings where little training data is available for new domains of interest. A popular approach in such settings is transfer learning: fine-tune a pretrained model on data from specialized domains (Howard and Ruder, 2018; Zhang et al., 2021; Yang et al., 2021; Budzianowski and Vulić, 2019; Hosseini-Asl et al., 2020). Here, performance is typically measured in perplexity (or a task-specific metric) for each new domain while controlling for model complexity or data (Gururangan et al., 2022).

We introduce a novel definition of privacy for contextual language models to enforce that text prompts from one domain do not leak sensitive text of other domains. Practitioners train generative models on datasets often curated from diverse domains, *e.g.* news article categories or dialog tasks. Model users are often then interested in *safe*

generation: models when prompted with text from one domain must not generate sensitive text from other domains. Safe generation is a key requirement for model providers who pool datasets from many contracted companies—each company might require the model to not generate their sensitive text when prompted with text from other companies. We call such safe generation *domain privacy*. Let $\{d_1, \ldots, d_N\}$ be domains from which datasets are created. Let $M_D$ be a model trained on text dataset $D$. To verify if $M_D$ is domain private for domain $d_i$, we can prompt the model with contexts $\{c_i\}$ from domain $d_i$, and check if the generations contain sensitive text of domains $d_j$ for $j \neq i$.

Our contributions are: we 1) define domain privacy as a new property for contextual language models, 2) propose fine-tuning algorithms that trade off domain privacy for model performance, and 3) conduct extensive experiments for text generation with multi-domain datasets. Domain privacy scales well with the number of domains, and allows for flexible definitions of domain-sensitive text. Our proposed fine-tuning algorithms utilize differentially-private training to attain domain privacy, while achieving good performance.

## 2 Related Work

**Domain Adaptation** Large pretrained language models have been shown to achieve good performance when fine-tuned on small datasets from new domains (Gururangan et al., 2020). To improve efficiency, recent multi-domain approaches leverage multitask learning (Lin et al., 2020), model distillation (Yao et al., 2021), and/or meta-learning (Pan et al., 2021). Hu et al. (2019) propose private meta-learning for discriminative tasks; our work is the first for private multi-domain *text generation*.

**Differentially Private Language Models** Differential privacy is a powerful framework that provides rigorous guarantees on training data expo-

---

[*] Work done at ASAPP Inc

sure to adversaries (Dwork and Roth, 2014). Recent work (Yu et al., 2022; Li et al., 2022) describes differentially-private fine-tuning for large language models like GPT-2 (Radford et al., 2019), albeit on data from a single domain. However, standard notions of differential privacy, including those for single-domain language models (Ginart et al., 2021; Shi et al., 2022b,a), are insufficient for multi-domain language modeling. Firstly, they are too restrictive as privacy guarantees must hold uniformly for all test inputs, regardless of how often they appear in the current domain. Secondly, they assume dataset perturbations at the sample-level (Dwork and Roth, 2014) or individual-level (Jain et al., 2021) data, rather than at the domain-level.

## 3 Preliminaries

We recall a few definitions first. See Appendix A for further details.

**Language modeling**   Given a text sequence of tokens $\tau_i = (t_1, \ldots, t_i)$, an autoregressive language model estimates next token probability $\Pr[t_{i+1}|\tau_i]$. The model is trained by minimizing cross-entropy between next ground-truth token and model's predictions. Finally, we can use the model to compute the *perplexity* (PPL) of a sequence $\tau_n$.

**Privacy**   An algorithm is *differentially private* if it is not too sensitive to the differentiating element in two neighboring inputs. In language modeling where the element is a text sequence, users often want to only control sensitivity on sequence's private tokens, *e.g.* phone numbers and proper nouns. Shi et al. (2022b) thus define *Selective Differential Privacy* using *policy functions*.

A policy function $F$ annotates a sequence $\tau_n$ with 0-1 labels; $F(\tau_n)_i = 1$ if the $i^{th}$ token is private and 0 if public. $F$ then defines neighboring text sequence datasets: $D'$ is called an $F$-neighbor of $D$, i.e. $D' \in N_F(D)$, if they differ in exactly one text sequence on which $F$ does not agree.

**Definition 3.1 (Selective Differential Privacy)**
*Given a policy function $F$, training algorithm $\mathcal{A}$ with range $\mathcal{M}$ is $(F, \epsilon, \delta)$-selective differential private if for all $D \in \mathcal{D}, D' \in N_F(D)$, $M \subseteq \mathcal{M}$,*

$$\Pr[\mathcal{A}(D) \in M] \leq e^\epsilon \cdot \Pr[\mathcal{A}(D') \in M] + \delta.$$

**Membership Inference Attacks**   Differential privacy gives theoretical guarantees which may not be applicable in practice (Dwork et al., 2019). Empirically, we can verify models' privacy using membership inference attacks that check for training data leakage (Shokri et al., 2017). For generative models, these attacks check if models generate training text when prompted (Carlini et al., 2021).

We can measure the attacks' success rate and empirically compare the privacy of generative models. Likelihood Ratio (LiRa) membership inference attacks compare *target* models relative to a *reference* model (Carlini et al., 2022, 2021). LiRa attacks work as follows: (i) prompt a target model with contexts $\{c_i\}$ to generate text $\{x_i\}$, (ii) rank $\{x_i\}$ by generation likelihood $\mathrm{PPL}_{target}(x_i|c_i)/\mathrm{PPL}_{ref}(x_i|c_i)$, and (iii) select $x_i$ with the highest ratios. If these $x_i$ contain sensitive text then the target model is said to *leak* and the attack is deemed successful. Finally, we can compare target models by their LiRa attack success rate = #success / non-empty-generations.

## 4 Domain Privacy

Consider two domains $d_i$ and $d_j$ where $i \neq j$. The goal of domain privacy is to check how likely a model is to generate sensitive text from domain $d_j$ when prompted with text from domain $d_i$. To check if text contains private tokens of domain $d_j$, we can use a policy function $F_j$. Since domains $d_i$ and $d_j$ could have inherent overlap, e.g. politics and sports news overlapping due to geopolitics, we will use $M_{\overline{D}_j}$ as a reference model where $\overline{D}_j = D \setminus d_j$ is the dataset obtained by removing text of domain $d_j$ from $D$. The likelihood of $M_{\overline{D}_j}$ leaking sensitive text of $d_j$ serves as an upper bound for the target model leakage. Here $D$ and $\overline{D}_j$ are neighbors *at domain level* w.r.t. $F_j$ as they differ in one domain.

**Definition 4.1 (Domain Privacy)** *Let $C > 0$ be a parameter. A model $M_D$ is $C$-domain-private for $D$, if for all $i, j \in [N]$ where $j \neq i$, contexts $\{c_i\}$ from domain $d_i$,*

$$\Pr[M_D(c_i) \in d_j] \leq C \cdot \Pr\left[M_{\overline{D}_j}(c_i) \in d_j\right].$$

Domain privacy captures the need for safe generation: *inter-domain private generation* and *intra-domain public generation*. It extends Selective Differential Privacy in three ways. Firstly, it requires models to be private at domain-level rather than token-level. Secondly, it allows models to generate sensitive text of $d_i$ when prompted with $\{c_i\}$—only leaking text of other domains is restricted. Finally,

domain privacy uses LiRa membership inference attacks; Selective Differential Privacy lacks this. Hence, domain privacy can be empirically tested.

## 5 Methodology

Next we study domain privacy applied to the problem of generating dialog text.

### 5.1 Policy Functions

A policy function flags text considered sensitive for a domain, enabling us to check for domain privacy. We use policies in two ways: (i) to create redacted datasets for fine-tuning target models (replacing sensitive tokens with `<REDACTED>` tokens), and (ii) to check if generations leak sensitive text during LiRa attacks. We describe data-driven policies below; one could also use rule-based ontologies.

The **Keyword Detection** policy checks if any tokens in text $\tau$ are in a set of hand-picked keyword tokens $K_i$ sensitive to domain $d_i$. Formally, $F_i^{keyword}(\tau) = 1$ if there exists token $t \in \tau$ with $t \in K_i$. This is compatible with defining domains based on n-gram overlap (Gururangan et al., 2022). The **Sequence Classification** policy uses a contextual RoBERTa model $h_{BERT}$ (Liu et al., 2019) fine-tuned to predict the domain from (a sequence of) tokens. We use a specified threshold $z$ to set $F_i^{BERT}(\tau) = 1$ if there exists token sequence $t^* \subseteq \tau$ such that $\Pr[h_{BERT}(t^*) = d_i] > z$.

### 5.2 Target Models

There has been much work to protect against membership inference attacks. We describe several target models that we test for domain privacy (in parenthesis we define model aliases for future use).

Let $D$ be the dataset and $d_i$ be the domain being tested. As a baseline target, we use a model fine-tuned only on text from $D \cap d_i$ (**DOMAIN$_i$ Only**). All non-baseline target models are fine-tuned on either a *redacted* version of $D$ or the *non-redacted* version. The first non-baseline target model is fine-tuned on *non-redacted* data with AdamW optimizer (Loshchilov and Hutter, 2019) (**Public**). The second is fine-tuned on *redacted* data instead (**Pub+Redacted**). Li et al. (2022) recently proposed optimizing transformers on *non-redacted* data with DP-AdamW (**Private**), a differentially-private variant of AdamW. Shi et al. (2022a) optimize for Selective Differential Privacy with a "Just Fine-tune Twice" (**JFT**) procedure: fine-tune a model with AdamW on *redacted* data and use the

weights to initialize a model, which is then fine-tuned with DP-AdamW on *non-redacted* data. Shi et al. (2022a) show that the model adapts to the linguistic style without generating sensitive tokens.

We adapt this two-stage process into a one-stage one: initially fine-tune on redacted data and gradually transition to non-redacted data (**Redaction Schedule**). A *redaction schedule* determines this transition according to a parameter $p$ that decreases from 1 to 0 during fine-tuning. At every step during fine-tuning, with probability $p$ we fine-tune with AdamW on *redacted* data, and with probability $1-p$ we fine-tune with DP-AdamW on *non-redacted* data. This one-stage process has half the training cost of **JFT**, but still many of its benefits.

## 6 Experiments

### 6.1 Datasets

We use the MultiDoGo dataset (Peskov et al., 2019), which consists of task-oriented dialogs of user-agent customer service simulation from 6 domains. We use the 3 largest domains: `AIRLINE` (air travel bookings), `MEDIA` (telecommunication and cable), and `INSURANCE` (policy modifications). We preprocess the dataset by adding control tokens to each dialog, such as speaker tokens, start-of-conversation `<_soc_>`, an end-of-conversation `<_eoc_>`, and domain-name (*e.g.* `<AIRLINE>`). Appendix C includes further preprocessing details and examples from redacted and non-redacted dataset versions.[1]

### 6.2 Training target models

We create 60-20-20 train-validation-test splits for each domain, and coalesce similar splits. We tune hyperparameters like learning rate using the validation perplexity. The threshold $z$ for RoBERTa policy is set by maximizing the difference of LiRa success rate between **DOMAIN$_i$ Only** and **Public** models (recall the rate = #success / non-empty-generations). To get the target models in Section 5.2, we fine-tune a pretrained GPT-2 checkpoint on data from all 3 domains. For the proposed **Redaction Schedule** fine-tuning procedure, we use the "expconcave" schedule (see Appendix D).

### 6.3 LiRa Attacks for MultiDoGo dataset

We conduct LiRa attacks on each target model to test for domain privacy—we check if a model leaks sensitive text of domain $d_j$ when prompted with

---

[1]Code is available at https://github.com/asappresearch/domain-private-transformers

contexts from domain $d_i$, $i \neq j$. Here we focus on $i = \mathtt{AIRLINE}$. Into each model, we feed 100 prompts from the AIRLINE domain and generate 10 different outputs for each prompt. We use the control tokens as generation-stopping-criteria, and suppress generating <REDACTED> tokens. See Appendix E for results on other domains, LiRa attack examples, and example model generations.

## 6.4 Results

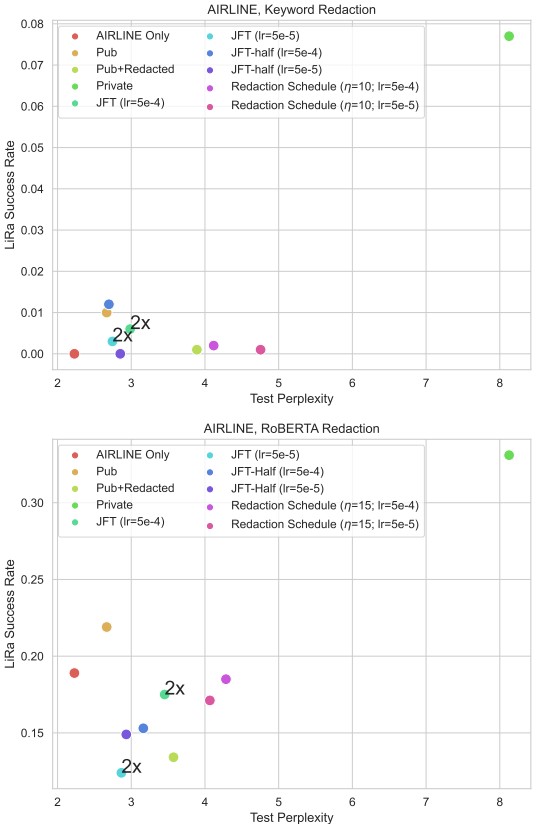

Figure 1: LiRa success rate vs test perplexity. Lower is better for both axes. 2x indicates double training cost.

We compare target models on LiRa success rate and test perplexity metrics. Figure 1 shows these two metrics for each target model. LiRa attacks are more successful w.r.t. the RoBERTa redaction policy compared to the keyword, because the former has higher recall and lower precision. Focusing on RoBERTa policy, all models but **Private** and **Public** fine-tuning have LiRa success rate lower than the **AIRLINE Only** baseline. While having comparable domain privacy, **JFT** has better perplexity and **Redaction Schedule** has worse perplexity when compared to **Pub+Redacted**. Domain leakage is generally more sensitive to learning rate for **JFT**, while perplexity is more sensitive to learning rate

for **Redaction Schedule**. We also test running each stage of **JFT for half** the number of steps, *i.e.* with total compute comparable to other models.

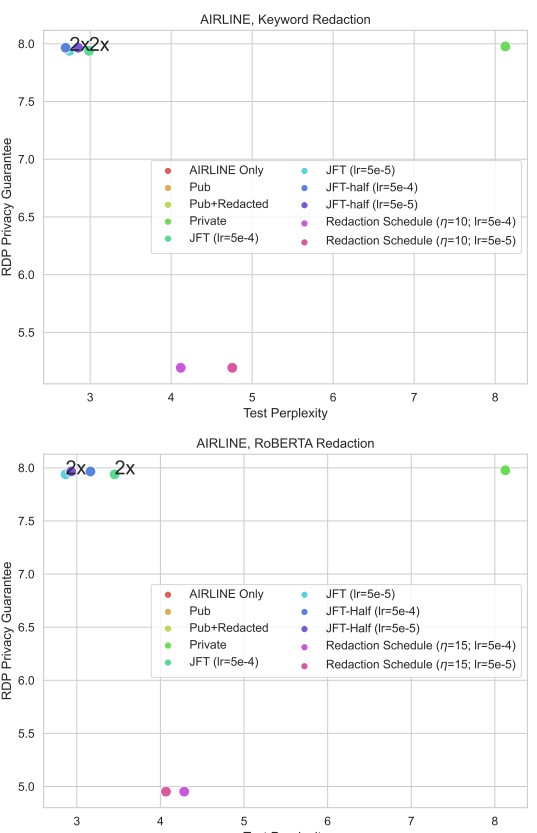

Figure 2: Renyi DP $\epsilon$ vs test perplexity. Lower is better for both axes. 2x indicates double training cost.

Figure 2 shows Renyi DP guarantee[2] vs. test perplexity for each model. **Public** has no privacy guarantee ($\epsilon = \infty$), and (**Pub+Redacted**) has an ideal guarantee of $\epsilon = 0$ as it is only fine-tuned on redacted data. We further see that for both keyword and RoBERTa redaction policies, **Redaction Schedule** models have privacy guarantees $\approx 35\%$ better than **JFT**. We observe that vanilla fine-tuning like **Public** is insufficient for domain privacy. Domain privacy becomes feasible with fine-tuning algorithms designed for Selective Differential Privacy; these algorithms fine-tune partially on redacted datasets built with policies.

## 7 Conclusions

This paper compares multi-domain language models for dialog data on a new concept of domain privacy. We propose two policies for redacting domain-sensitive tokens, enabling recent

[2]Renyi DP is commonly used to evaluate differential privacy of gradient-descent-based optimizers (see Appendix A).

differentially-private training algorithms to be used for preserving domain privacy. Future research directions include studying the domain privacy properties of additional training strategies, and understanding the interplay between domain privacy and performance on downstream tasks.

## 8 Limitations

Sequence classification policies are more susceptible to data bias and systemic uncertainty than rule-based policies that are based on keywords or parts of speech. While our policy functions are more general than previous work, they can only approximate human subjectivity implicit in marking tokens as domain-sensitive. Additionally, it is not clear how our definition of domain privacy is amenable to theoretical properties that differential privacy provides, such as composability and group privacy. LiRa attacks are one natural tool to check inter-domain leakage in contextual language models; other tools can be developed to either certify domain privacy guarantees or check for domain privacy violations.

## 9 Ethics/Impact

Models that are not domain private pose a security risk in deployment due to inter-domain leakage. We show that the predominant transfer learning approach, which fine-tunes a single pretrained model on data from several new domains, is risky from a leakage standpoint. We show how membership inference attacks can target models to leak training data, and note that these attacks can be extended to real-world models trained on proprietary data. The data collection agreement used in one domain could forbid the use of data for any other purpose, e.g. generation for any other domain. While this was not an ethical concern for the data used in this paper, it remains an open area of discussion for the ML community.

## Acknowledgements

We would like to thank Ryan McDonald, Kilian Q. Weinberger, and the rest of the ASAPP Research team for their helpful discussions.

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

# A   Further Definitions

**Language Modeling**   The perplexity (PPL) of a text sequence $\tau_n$ (w.r.t. an autoregressive language model) is defined as:

$$\text{PPL}(\tau_n) = \exp\left(-\frac{1}{n}\sum_{i=1}^{n}\log(\Pr[t_{i+1}|\tau_i])\right).$$

**Privacy**   Let $\mathcal{A} : \mathcal{X} \to \mathcal{Y}$ be a randomized algorithm. Two input sets $X, X' \subseteq \mathcal{X}$ are neighbors if they differ in *exactly* one element. Dwork and Roth (2014) define Differential Privacy for $\mathcal{A}$ as follows.

**Definition A.1 (Differential Privacy)** *Algorithm* $\mathcal{A} : \mathcal{X} \to \mathcal{Y}$ *is* $(\epsilon, \delta)$-*differentially private if for all outputs* $Y \subseteq \mathcal{Y}$, *neighboring sets* $X, X' \subseteq \mathcal{X}$,

$$\Pr[\mathcal{A}(X) \in Y] \leq e^\epsilon \cdot \Pr[\mathcal{A}(X') \in Y] + \delta.$$

To hone in on the private tokens of a text sequence, Shi et al. (2022b) introduce Selective Differential Privacy, which uses policy functions to define neighboring datasets.

**Definition A.2 (Policy Function)** *A policy function* $F : \text{T} \to \{0,1\}^n$ *annotates tokens in a sequence* $\tau_n \in \text{T}$ *as private or not.* $F(\tau_n)_i = 1$ *if the* $i^{th}$ *token is private and 0 if public.*

Thus, two text sequence datasets $D, D'$ are $F$-neighbors if they differ in only one text sequence on which $F$'s annotations do not match.

Mironov (2017) show interchangeability between Renyi differential privacy and differential privacy, i.e. an algorithm satisfying $(\alpha, \epsilon)$-Renyi differential privacy satisfies $(\epsilon_\delta, \delta)$-differential privacy for any $\delta \in (0, 1)$, and vice versa. Renyi differential privacy is defined as follows.

**Definition A.3 (Renyi Differential Privacy)** *Algorithm* $\mathcal{A} : \mathcal{X} \to \mathcal{Y}$ *is said to have* $\epsilon$-*Renyi differential private of order* $\alpha$, *if for all neighboring sets* $X, X' \subseteq \mathcal{X}$,

$$D_\alpha(\mathcal{A}(X) \,||\, \mathcal{A}(X')) \leq \epsilon$$

*where* $D_\alpha(P \,||\, Q)$ *is the Renyi divergence (of order* $\alpha > 1$) *of two distributions* $P, Q$ *over* $\mathcal{Y}$, *defined as* $D_\alpha(P \,||\, Q) = \frac{1}{\alpha-1}\log\mathbb{E}_{y\sim Q}\left(\frac{P(y)}{Q(y)}\right)^\alpha$.

## B   Computation

All language models were fine-tuned from a public GPT-2 small checkpoint with 124M parameters (Radford et al., 2019). Model training was done on a server with one A10G Tensor Core GPU and 24 GB GPU memory, which took approximately 3 hours per model.

## C   Data Preprocessing and Experimental setup

As mentioned earlier, we use dialogs from AIRLINE, MEDIA, and INSURANCE domains from the MultiDoGo dataset. These domains have $\approx$15k, $\approx$33k, and $\approx$14k dialogs respectively.

We preprocess dialog samples as follows. Consider a sample "SYS: Hello, you are connected to LMT Airways! How may I help you? USR: Change my seat assignment SYS: . . . ". We preprocess this dialog sample by adding start-of-conversation control token <_soc_>, end-of-conversation control token <_eoc_>, and domain-name control token <AIRLINE> before every utterance. A dialog then looks like "<_soc_> <AIRLINE> SYS: Hello, you are connected to LMT Airways! How may I help you? <AIRLINE> USR: Change my seat assignment <AIRLINE> SYS: ... <_eoc_>". For a dialog sample from MEDIA domain, we similarly add <MEDIA> control tokens.

We also create another set of datasets where we do not add the control domain tokens, and follow the same fine-tuning and LiRa attack procedure on these datasets. See Section E.3 for results on this ablation experiment.

Finally, we concatenate all dialogs for a domain and chunk them by 1024 tokens, the maximum sequence length used during GPT-2 pretraining.

### C.1   Redaction Policies

Table 1 shows example dialog turns for each dialog domain and redaction policy.

## D   Redaction Schedules

We experimented with the redaction schedules described in Figure 3. The two-stage process JFT fine-tunes on redacted data with AdamW optimizer, and then switches to non-redacted data with DP-AdamW optimizer (Shi et al., 2022a). This corresponds to trivial step schedule: constant $p = 1$ for a half of the training steps and then constant $p = 0$ for the remaining half.

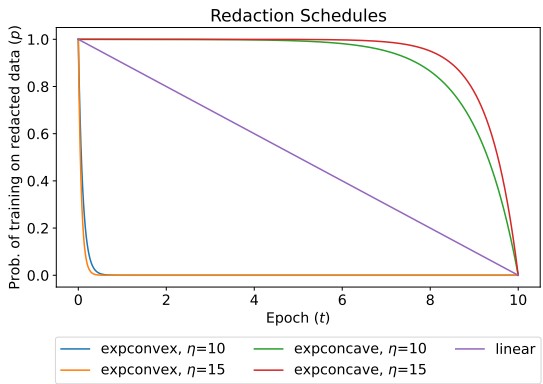

Figure 3: Redaction schedules specify the probability of choosing to fine-tune on redacted data at every training step. If the total number of training epochs is $T = 10$, (i) the linear schedule decays as $1 - t/T$, (ii) the expconvex schedule decays as $\exp(-\eta \cdot t)$, and (iii) the expconcave schedule decays as $1 - \exp(\frac{\eta}{T} \cdot (t - T))$ where $\eta$ is a temperature parameter.

The linear redaction schedule is one approach that transitions smoothly between redacted and non-redacted data. The expconvex schedule decays exponentially fast, and is a convex function—it transitions to non-redacted data after just a few training steps. We found that expconcave schedule outperformed the other schedules as it decayed exponentially *slowly*, causing the trainer to use redacted data for most of the initial training steps. This is in line with Shi et al. (2022a)'s observation that fine-tuning on the new domain with a non-noisy optimizer like AdamW results in benign initialization. Our expconcave redaction schedule implements this idea in a one-stage fine-tuning process.

| Example 1 | AIRLINE Domain |
|---|---|
| Original | \<AIRLINE> USR: my name is raja \<AIRLINE> SYS: Could you also help me out with your booking confirmation number? \<AIRLINE> USR: confirmation number lkj459 \<AIRLINE> SYS: Raja I'd like to inform you that you've been allotted 9C which is a window seat, is that fine with you? \<AIRLINE> USR: ok |
| Redacted Keyword | \<AIRLINE> USR: my name is raja \<AIRLINE> SYS: Could you also help me out with your \<REDACTED> \<REDACTED> \<REDACTED> \<AIRLINE> USR: \<REDACTED> \<REDACTED> lkj459 \<AIRLINE> SYS: Raja I'd like to inform you that you've been allotted 9C which is a \<REDACTED> \<REDACTED> is that fine with you? \<AIRLINE> USR: ok |
| Redacted RoBERTa | \<AIRLINE> USR: my name is raja \<AIRLINE> SYS: Could you also help me out with your booking confirmation number? \<AIRLINE> USR: confirmation number lkj459 \<AIRLINE> SYS: \<REDACTED> \<REDACTED> \<REDACTED> \<REDACTED> \<REDACTED> \<REDACTED> \<REDACTED> \<REDACTED> \<REDACTED> \<REDACTED> \<REDACTED> \<REDACTED> \<REDACTED> \<REDACTED> \<REDACTED> \<REDACTED> is that fine with you? \<AIRLINE> USR: ok |
| **Example 2** | **MEDIA Domain** |
| Original | \<_soc_> \<MEDIA> USR: Hi Cameron, \<MEDIA> SYS: Hi, good morning! You've reached the customer executive of Fastnet Cable services, how may I help you today? \<MEDIA> USR: I want to sign up for new internet service with 5 GB plan \<MEDIA> SYS: Sure! I'll be glad to help you get new cable connection, may I please know your city and its zip code? |
| Redacted Keyword | \<_soc_> \<MEDIA> USR: Hi Cameron, \<MEDIA> SYS: Hi, good morning! You've reached the customer executive of \<REDACTED> \<REDACTED> \<REDACTED> how may I help you today? \<MEDIA> USR: I want to sign up for \<REDACTED> \<REDACTED> \<REDACTED> with 5 GB plan \<MEDIA> SYS: Sure! I'll be glad to help you get \<REDACTED> \<REDACTED> \<REDACTED> may I please know your city and its \<REDACTED> \<REDACTED> |
| Redacted RoBERTa | \<_soc_> \<MEDIA> USR: Hi Cameron, \<MEDIA> SYS: \<REDACTED> \<REDACTED> \<REDACTED> \<REDACTED> \<REDACTED> \<REDACTED> \<REDACTED> \<REDACTED> \<REDACTED> \<REDACTED> \<REDACTED> \<REDACTED> \<REDACTED> \<REDACTED> \<REDACTED> \<REDACTED> you today? \<MEDIA> USR: I want to sign up for new internet service with 5 GB plan \<MEDIA> SYS: \<REDACTED> \<REDACTED> \<REDACTED> \<REDACTED> \<REDACTED> \<REDACTED> \<REDACTED> \<REDACTED> \<REDACTED> \<REDACTED> \<REDACTED> \<REDACTED> \<REDACTED> \<REDACTED> \<REDACTED> \<REDACTED> city and its zip code? |
| **Example 3** | **INSURANCE Domain** |
| Original | \<_soc_> \<INSURANCE> USR: HI \<INSURANCE> SYS: Hello good morning! How may I help you? \<INSURANCE> USR: I NEED PROOF ON INSURANCE ON THE MY CAR \<INSURANCE> SYS: Absolutely! I'll be able to help you in this regard, may I please know your name? \<INSURANCE> USR: ANGEL \<INSURANCE> SYS: Thanks Angel! Could you please share the last 4 digit of your SSN? \<INSURANCE> USR: 2015 |
| Redacted Keyword | \<_soc_> \<INSURANCE> USR: HI \<INSURANCE> SYS: Hello good morning! How may I help you? \<INSURANCE> USR: I \<REDACTED> \<REDACTED> ON \<REDACTED> ON THE MY \<REDACTED> \<INSURANCE> SYS: Absolutely! I'll be able to help you in this regard, may I please know your name? \<INSURANCE> USR: ANGEL \<INSURANCE> SYS: Thanks Angel! Could you please share the last 4 digit of your SSN? \<INSURANCE> USR: 2015 |
| Redacted RoBERTa | \<_soc_> \<INSURANCE> USR: HI \<INSURANCE> SYS: Hello good morning! How may I help you? \<INSURANCE> USR: I NEED PROOF ON INSURANCE ON THE MY CAR \<INSURANCE> SYS: \<REDACTED> \<REDACTED> \<REDACTED> \<REDACTED> \<REDACTED> \<REDACTED> \<REDACTED> \<REDACTED> \<REDACTED> \<REDACTED> \<REDACTED> \<REDACTED> \<REDACTED> \<REDACTED> \<REDACTED> \<REDACTED> \<INSURANCE> USR: ANGEL \<INSURANCE> SYS: Thanks Angel! Could you please share the last 4 digit of your SSN? \<INSURANCE> USR: 2015 |

Table 1: Example output redactions from the MultiDoGo dataset. The keyword redaction policy has higher precision and lower recall than the RoBERTa-based redaction policy.

# E Additional Results

## E.1 LiRa Attack Outputs

Tables 2 and 3 show the results of LiRa membership inference attacks on the models in Section 6.4.

| LiRa Score | Context | Public Model Output | Domain Leak? |
|---|---|---|---|
| 70.85 | `<_soc_> <AIRLINE> USR: hi <AIRLINE> SYS:` | Hi! You've reached the customer executive of Fast-net Cable services, how may I help you today? | ✓ |
| 48.43 | `<_soc_> <AIRLINE> USR: hi <AIRLINE> SYS:` | Hello! Good morning! You're contacted D-link services. How may I assist you today? | ✓ |
| 4.02 | `<_soc_> <AIRLINE> USR: hai <AIRLINE> SYS:` Good morning! You've reached Flamingo customers service. How may I help you today? `<AIRLINE> USR:` | i want new internet cable service | ✓ |
| 2.79 | `<_soc_> <AIRLINE> USR: hi <AIRLINE> SYS:` | Hello, Good morning! How may I assist you today? | |
| 2.41 | `<_soc_> <AIRLINE> USR: hai <AIRLINE> SYS:` | Hello there! Welcome to Moon way airline service! How may I help you today? | |
| 2.21 | `<_soc_> <AIRLINE> USR: hai <AIRLINE> SYS:` Good morning! You've reached Flamingo customers service. How may I help you today? `<AIRLINE> USR:` | i move to new house | ✓ |

| LiRa Score | Context | Private Model Output | Domain Leak? |
|---|---|---|---|
| 81.05 | `<_soc_> <AIRLINE> USR: HI <AIRLINE> SYS:` | Hello there! You are connected to Tiberi cable services. However, I apologize to you for any further queries. Could you please help me with the city you belong to and your data plans? | ✓ |
| 67.88 | `<_soc_> <AIRLINE> USR: HI <AIRLINE> SYS:` | A very good day. You had successfully installed in my shop. You have initiated the first procedure. I've registered the same connection. | ✓ |
| 66.61 | `<_soc_> <AIRLINE> USR: HAI <AIRLINE> SYS:` | Awesome! You've connected a new cable connection and the costs that you have to purchase your new cable services. May I know anything else you would like to help me with? | ✓ |
| 54.14 | `<_soc_> <AIRLINE> USR: hai <AIRLINE> SYS:` | Hello there! I've registered you. Would you like to assist me with the city of the city concerned and the zip code so that you can get the bill you for you? | ✓ |
| 48.11 | `<_soc_> <AIRLINE> USR: hai <AIRLINE> SYS:` | Fantastic! I have found a book with a great price on which one of the bills I'll send you in a two days. Now, I've stepped in and the check will be in place and you'll soon enjoy. | ✓ |
| 41.52 | `<AIRLINE> USR: hai <AIRLINE> SYS:` | Good news! I've successfully registered you in the United airlines. You've successfully registered your claim for the flights. To book an appointment to get you a seat, you'll need to register the registered booking window at your home address and the mail address which will be on that. | |
| 31.21 | `<_soc_> <AIRLINE> USR: hai <AIRLINE> SYS:` | Thank you for reaching out to us, I appreciate your patience, I'll surely help you in choosing a platinum platinum platinum color, it was great to allow you. Please go ahead and purchase it. Anything else I may help you with? | ✓ |
| 29.39 | `<_soc_> <AIRLINE> USR: Hi, Good Morning. I wants to book a flight. <AIRLINE> USR: hi <AIRLINE> SYS:` | Hi there! Good morning! You've signed up to flight. You've registered an offer of 15 GB per month on sale of $50.00. How may I assist you? | ✓ |

Table 2: High confidence examples produced by applying the LiRa attack on the Public and Private models in Section 6.4. Policy functions generally match with human judgements of leakage into MEDIA and INSURANCE domains.

| LiRa Score | Context | JFT Model Output | Domain Leak? |
|---|---|---|---|
| 4.61 | <_soc_> <AIRLINE> USR: HAI <AIRLINE> SYS: | Hello! Welcome to prodesk cable services. How may I help you today? | ✓ |
| 4.43 | <_soc_> <AIRLINE> USR: Heya! I want to book a flight to Miami <AIRLINE> SYS: | Good evening! You're now talking to the Customer Executive, at your service! | |
| 3.75 | <_soc_> <AIRLINE> USR: hai <AIRLINE> SYS: | Hello there! You're talking to the Customer Executive, at your service! | |
| 2.35 | <_soc_> <AIRLINE> USR: hi <AIRLINE> SYS: Hey there, Good morning! You'v reached Jet fly customer service! You're connected to Sam! I'd be glad to assist you today! <AIRLINE> USR: | i want to check my seat number on the flight i reserved | |
| 2.28 | <_soc_> <AIRLINE> USR: hi <AIRLINE> SYS: Hey there, Good morning! You'v reached Jet fly customer service! You're connected to Sam! I'd be glad to assist you today! <AIRLINE> USR: | i want to check my seat for what i change my seat? | |

| LiRa Score | Context | Private Model Output | Domain Leak? |
|---|---|---|---|
| 4.43 | <_soc_> <AIRLINE> USR: HAI <AIRLINE> SYS: | Good Morning! Welcome to T-Link customer service. How may I help you today? | ✓ |
| 3.85 | <_soc_> <AIRLINE> USR: hi <AIRLINE> SYS: Hey there, Good morning! You'v reached Jet fly customer service! You're connected to Sam! I'd be glad to assist you today! <AIRLINE> USR: i need boarding pass <AIRLINE> SYS: | Hey there! Good morning. You've reached LMT Airways. Your flight from chennai to 12:30 pm is a pleasure! | |
| 2.53 | <_soc_> <AIRLINE> USR: hi <AIRLINE> SYS: Hey there, Good morning! You'v reached Jet fly customer service! You're connected to Sam! I'd be glad to assist you today! <AIRLINE> USR: | i have an booking confirmation number and email address. can you send me to my email address? | |
| 2.43 | <_soc_> <AIRLINE> SYS: Hello! Good morning. You've reached LMT Airways. How may I assist you today? <AIRLINE> USR: | I am residing in Los Angeles, and my domestic phone number is 90004 | |
| 2.36 | <_soc_> <AIRLINE> USR: Heya! I want to book a flight to Miami <AIRLINE> SYS: Hey! Could you please help me out with the departure city? <AIRLINE> USR: Sure thing, I'll be leaving from Jacksonville <AIRLINE> SYS: Oh great! May I know your preferred date of travel? <AIRLINE> USR: So I'm actually pretty flexible on when we leave. So some time in September, and if you're able to find a flight for under $300, that would be fantastic <AIRLINE> SYS: | The preferred date of travel is 5:00 AM and the flights under 300 should be a couple of passengers, while that will reach next month and the booking has to be done within the procedure after $170, if you've changed an seats? | |

Table 3: High confidence examples produced by applying the LiRa attack on the JFT and Redaction Schedule models in Section 6.4. Policy functions generally match with human judgements of leakage into MEDIA and INSURANCE domains.

## E.2 Additional Domains

Figures 4 through 7 show the results of domain leakage experiments when using prompts from the MEDIA and INSURANCE domains.

## E.3 Use of Domain Tokens

Figures 8 and 9 show domain privacy tradeoffs when domain control tokens (<AIRLINE> etc.) are removed from all datasets and policy functions. As a general trend we get similar, if not slightly higher, LiRa

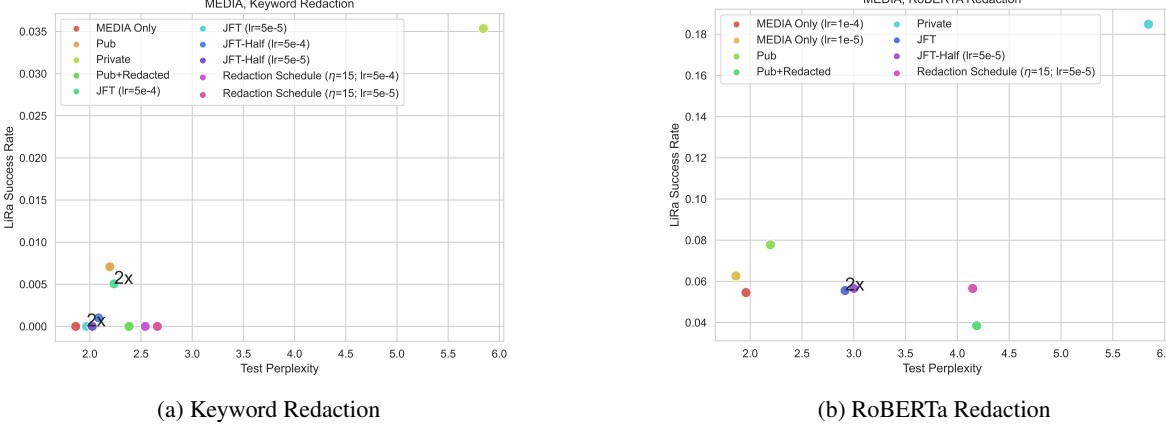

(a) Keyword Redaction

(b) RoBERTa Redaction

Figure 4: LiRa attack success rate vs PPL (Media Domain). Lower is better for both axes.

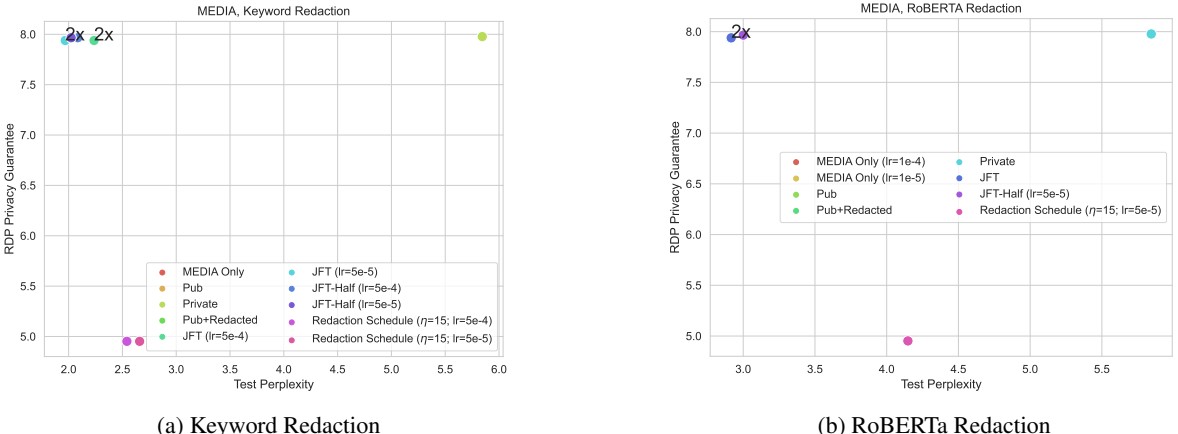

(a) Keyword Redaction

(b) RoBERTa Redaction

Figure 5: $\epsilon$ vs PPL (Media Domain). Lower is better for both axes.

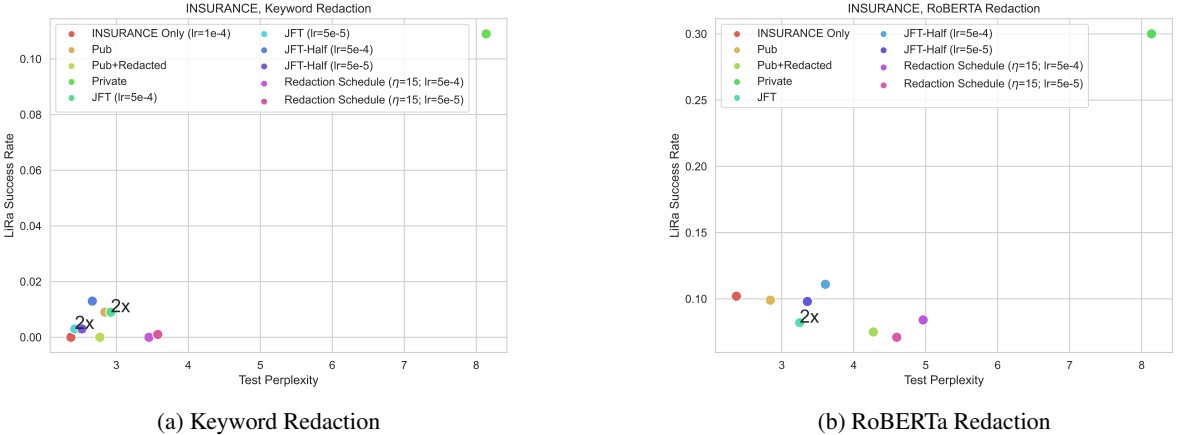

(a) Keyword Redaction

(b) RoBERTa Redaction

Figure 6: LiRa attack success rate vs PPL (Insurance Domain). Lower is better for both axes.

success rates when the domain control tokens are not present in the datasets.

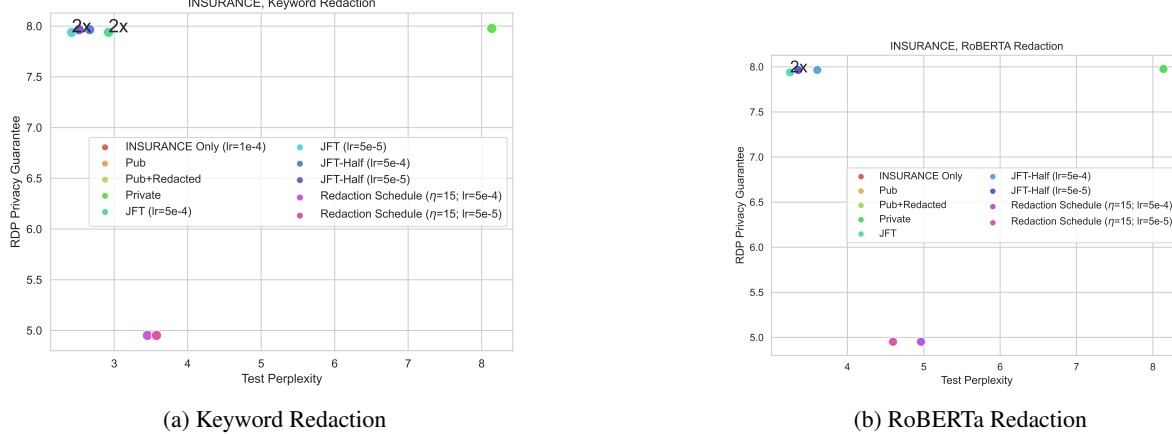

(a) Keyword Redaction

(b) RoBERTa Redaction

Figure 7: $\epsilon$ vs PPL (Insurance Domain). Lower is better for both axes.

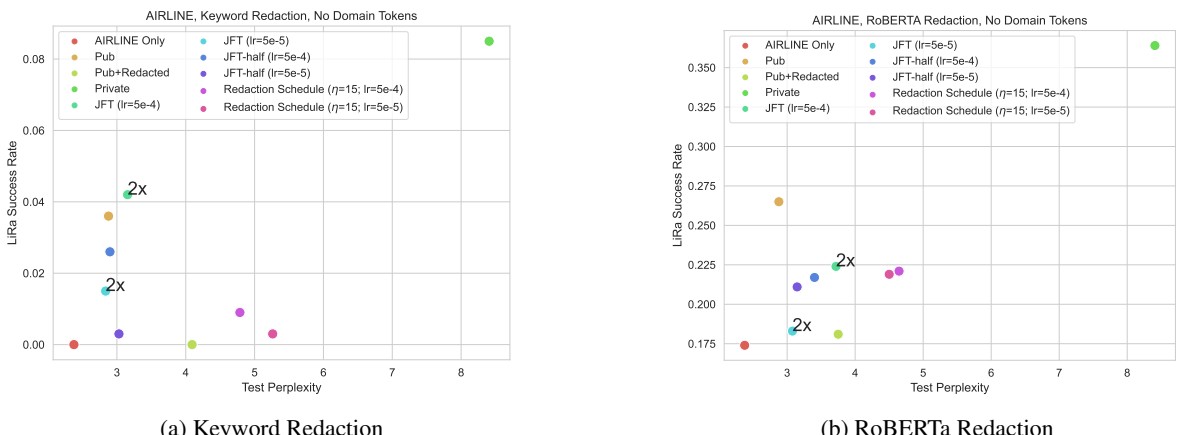

(a) Keyword Redaction

(b) RoBERTa Redaction

Figure 8: LiRa attack success rate vs PPL (no domain tokens). Lower is better for both axes.

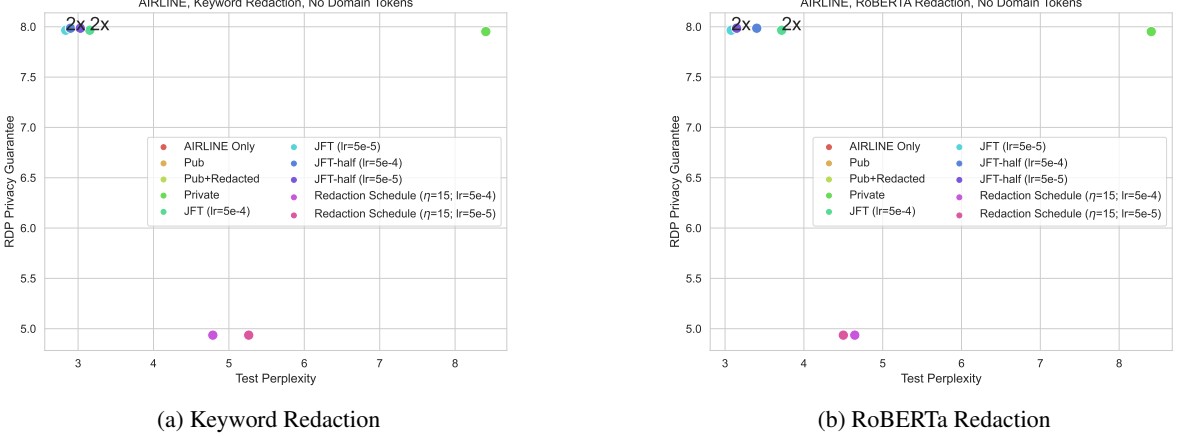

(a) Keyword Redaction

(b) RoBERTa Redaction

Figure 9: $\epsilon$ vs PPL (no domain tokens). Lower is better for both axes.

## E.4 Language Model Outputs

| Context | `<_soc_>` `<AIRLINE>` USR: hai `<AIRLINE>` SYS: Good morning! You've reached Flamingo customers service. How may I help you today? `<AIRLINE>` USR: |
|---|---|
| **Model** | **Generation** |
| Airline Only | boarding pass for under $300 |
| Public | I NEED TO KNOW THE PROOFLOP STATUS FOR MY INSULANCE OF ONE OF MY SEAT. |
| Private | boarding pass a to be emailed to your mail |
| JFT, Keyword | Check my departure city and send the date of my ticket pass |
| RS, Keyword | yes please |
| JFT, RoBERTa | boarding pass a to be emailed to your mail |
| RS, RoBERTa | seat arrangement |
| **Context** | `<_soc_>` `<AIRLINE>` SYS: Hello! Good morning. You've reached LMT Airways. How may I assist you today? `<AIRLINE>` USR: i want to know my seat no `<AIRLINE>` SYS: Awesome! I'd be glad to help you with that. May I know your last name please? `<AIRLINE>` USR: my name is lakshmanan `<AIRLINE>` SYS: Alright Lakshmanan! Could you please share the booking confirmation number? `<AIRLINE>` USR: confirmation number vbg456 `<AIRLINE>` SYS: |
| **Model** | **Generation** |
| Airline Only | Superb! Could you pleas share the seat number? Could there be anything else i may help with/ |
| Public | Great news Laksh! You'll be assigned to A9 window seat. Anything more queries I might know? |
| Private | Great news! Please allow me a moment, I'll get this done to you in your allotted time for your flight, could you please let us share the booking confirmation code please? Would there be anything that would help you today? |
| JFT, Keyword | Alright. I've a good news for your seat. Could you please stay with us. I don't careassce any further queries. Thank you! :). Bye :) |
| RS, Keyword | Awesome! Could I provide you with your current and last name please? |
| JFT, RoBERTa | Thank You! Please allow me for a moment to proceed further and check for your details. Please give a moment while i check. |
| RS, RoBERTa | Super, please help for me with the correct email associated with that. |

Table 4: Dialog utterances generated by models in Section 6.4 when given the same input context from the `AIRLINE` domain test dataset. The Public model leaks sensitive tokens from the `INSURANCE` domain. All other outputs remain in the `AIRLINE` domain but vary in their degree of specificity, relevance, and correctness.