# OpenReview forum: "Domain Private Transformers for Multi-Domain Dialog Systems"
_EMNLP/2023/Conference — EMNLP 2023 Findings_

### Official Review · Reviewer_tU8m · 2023-08-03

**Soundness:** 3

**Excitement:**

3: Ambivalent: It has merits (e.g., it reports state-of-the-art results, the idea is nice), but there are key weaknesses (e.g., it describes incremental work), and it can significantly benefit from another round of revision. However, I won't object to accepting it if my co-reviewers champion it.

**Paper Topic And Main Contributions:**

The paper defines domain privacy towards transformers, which is an important and novel property of differential privacy on language models. It further proposes a redaction schedule finetuning method to attain domain privacy as well as achieve good performance.

**Questions For The Authors:**

1. How to evaluate the leakage information or sensitive text as mentioned in line 140 when two different domains have common features? For example, booking tickets is a common feature in both the AIRLINE and MOVIE THEATER. Then there may be a same subset of target words in the two domains.
2. Given large language models are encoded with a large amount of knowledge and show decent reasoning abilities, will the definition and method still be useful for them?


**Reasons To Accept:**

1. The paper is well-written and easy to follow.
2. It defines an important property of private transformers. Extensive experiments and sufficient analysis are presented to support the main claims.


**Reasons To Reject:**

1. The definition and method may be hard to generalize to language models with larger size or more pretrained knowledge.

**Reproducibility:**

3: Could reproduce the results with some difficulty. The settings of parameters are underspecified or subjectively determined; the training/evaluation data are not widely available.

**Reviewer Confidence:**

2: Willing to defend my evaluation, but it is fairly likely that I missed some details, didn't understand some central points, or can't be sure about the novelty of the work.

---

> ### Author Rebuttal · Authors · 2023-08-28
>
> We thank the reviewer for their insights. We are pleased that they find the paper well-written, and that our experiments were extensive and sufficient. We respond to their comment and questions below:
>
> > The definition and method may be hard to generalize to language models with larger size or more pretrained knowledge.
>
> Our definition of domain privacy is flexible enough to generalize to LLMs. Different models/datasets/training algorithms would lead to different values of $C$ in domain privacy, as the models would differ in how much cross-domain sensitive information they leak when prompted with text. In fact, determining the $C$ privacy parameter for LLMs is interesting future work.
>
> Response to questions:
> > How to evaluate the leakage information or sensitive text as mentioned in line 140 when two different domains have common features? For example, booking tickets is a common feature in both the AIRLINE and MOVIE THEATER. Then there may be a same subset of target words in the two domains.
>
> With the AIRLINE/MOVIE THEATER example, the reviewer suggests good intuition for the domain privacy definition using policy functions, or policies. Introduced by [Shi et al. 2022b], policies for a domain mark the sensitive text tokens particular to that domain. This allows us to check for sensitive text in model generations. So, overlap in cross-domain sensitive information is exactly captured by policies.
>
> Definition 4.1 of domain privacy compares the model $M_D$ trained on the full dataset $D$ with reference models $M_{D \setminus d_j}$ trained on all data of $D$ except of domain $d_j$. On prompts of domain $d_i$ where $i \neq j$, if the reference model $M_{D \setminus d_j}$ leaks plentiful sensitive text of $d_j$, then domains $d_i$ and $d_j$ likely have large overlap. So, we are always comparing the domain privacy of $M_D$ against reference models that quantify cross-domain overlap. Thus, domain privacy also depends on the choice of policies, which determine what is considered sensitive in the first place.
>
> > Given large language models are encoded with a large amount of knowledge and show decent reasoning abilities, will the definition and method still be useful for them?
>
> Assuming open-access to LLMs, domain privacy is even more important for them since the training data is large and pooled from diverse sources. LLMs show decent zero-shot reasoning abilities, but they still can hallucinate text across domains. Our proposed methods use membership inference attacks, which require likelihoods of generated text. For closed-source models, this is an active area of research and interesting future work.
>
> [Shi et al. 2022b] [Selective differential privacy for language modeling](https://arxiv.org/abs/2108.12944). In NAACL.

---

### Official Review · Reviewer_sL5m · 2023-08-04

**Soundness:** 3

**Excitement:**

3: Ambivalent: It has merits (e.g., it reports state-of-the-art results, the idea is nice), but there are key weaknesses (e.g., it describes incremental work), and it can significantly benefit from another round of revision. However, I won't object to accepting it if my co-reviewers champion it.

**Paper Topic And Main Contributions:**

The paper discusses the performance of large, general-purpose language models in various conversational domains. While these models achieve low overall perplexity, there is a concern about their outputs staying within the intended domain. The paper introduces the concept of domain privacy to quantify the likelihood of a language model leaking across domains. It proposes policy functions based on token-level domain classification and an efficient fine-tuning method to enhance the model's domain privacy. Experimental results on membership inference attacks demonstrate that this method is comparable in resilience to existing differentially private language models.

**Reasons To Accept:**

1. The study of domain-level differential privacy for the Transformer model is novel and interesting.

2. The experimental analyses are intuitive and make sense.

**Reasons To Reject:**

1. The title of this paper seems confusing. It may claim that this work aims to improve the Tranfomer model, if it is the case, the authors should cover an extensive of tasks by Transformer, and the methodology should be specifically designed for the Transformer architecture. I think the motivation of this work is not clear.

2. It also lacks detailed discussions between this work and other work on the privacy of language models. This makes the contribution of this work weak to me.

**Reproducibility:**

3: Could reproduce the results with some difficulty. The settings of parameters are underspecified or subjectively determined; the training/evaluation data are not widely available.

**Reviewer Confidence:**

3: Pretty sure, but there's a chance I missed something. Although I have a good feel for this area in general, I did not carefully check the paper's details, e.g., the math, experimental design, or novelty.

---

> ### Author Rebuttal · Authors · 2023-08-28
>
> We thank the reviewer for their comments. We are delighted that they find our work novel and the experiments intuitive. Below we respond to their concerns:
>
> > The title of this paper seems confusing. It may claim that this work aims to improve the Tranfomer model, if it is the case, the authors should cover an extensive of tasks by Transformer, and the methodology should be specifically designed for the Transformer architecture.
>
> We agree with the reviewer that the title “Domain Private Transformers” may suggest a wider scope of tasks than our current results. We wanted to introduce the novel concept of domain privacy for large language models, where transformers are trained on data from multiple domains. Since our methods apply for transformers used in dialog systems, we can modify our title to “Domain Private Transformers for Multi-Domain Dialog Systems”. This would alleviate the reviewer’s concern.
>
> > I think the motivation of this work is not clear.
>
> We motivate domain privacy for contextual language models from real-world concerns of practitioners. Consider the following concrete example. To train a dialog generator, a model provider pools data from companies of different industries, such as an airline, a movie theater, and a bank. Each company would not want their industry-specific data to be generated when prompted with text particular to other industries. Our proposed domain privacy addresses this real “safe generation” requirement that many model providers face.
>
> > It also lacks detailed discussions between this work and other work on the privacy of language models. This makes the contribution of this work weak to me.
>
> Domain privacy draws from literature on differential privacy, domain adaptation, and adversarial text generation attacks such as membership inference attacks. We discuss this in Sections 2 (Related Work) and 3 (Preliminaries). We also build upon recent empirical work [Li et al. 2022, Shi et al. 2022a-b] on private fine-tuning, and compare our methods with them. Space constraints limit us from further discussing related work, but we’d be happy to add it in the appendix.
>
> [Li, et al. 2022] [Large language models can be strong differentially private learners](https://arxiv.org/abs/2110.05679). In ICLR.
>
> [Shi et al. 2022a] [Just fine-tune twice: Selective differential privacy for large language models](https://arxiv.org/abs/2204.07667). In EMNLP.
>
> [Shi et al. 2022b] [Selective differential privacy for language modeling](https://arxiv.org/abs/2108.12944). In NAACL.

---

### Official Review · Reviewer_nhsK · 2023-08-11

**Soundness:** 3

**Excitement:**

4: Strong: This paper deepens the understanding of some phenomenon or lowers the barriers to an existing research direction.

**Paper Topic And Main Contributions:**

The paper introduces "domain privacy" for contextual language models, ensuring that prompts from one domain do not produce sensitive outputs from other domains. It addresses concerns especially relevant for entities pooling data from multiple sources that require non-leakage of their sensitive information. The main contributions are:

* Definition of domain privacy for language models.
* Fine-tuning algorithms balancing domain privacy and model performance.
* Experiments on text generation with multi-domain datasets, demonstrating that the proposed techniques achieve domain privacy while maintaining performance.

**Reasons To Accept:**

*  I really like notion of Domain Privacy, especially the scenario presented: a vendor serving language models to multiple clients across various domains must prevent information leaks from one client's domain to another's.
*  Algorithm is simple, redact tokens from other domains.
* Good experiment setup using LiRA [Carlini 2021] as the test bed for privacy leakage.


Ref:

[Carlini 2021] Membership Inference Attacks From First Principles


**Reasons To Reject:**

Nothing I can think of.

**Reproducibility:**

3: Could reproduce the results with some difficulty. The settings of parameters are underspecified or subjectively determined; the training/evaluation data are not widely available.

**Reviewer Confidence:**

4: Quite sure. I tried to check the important points carefully. It's unlikely, though conceivable, that I missed something that should affect my ratings.

---

> ### Author Rebuttal · Authors · 2023-08-28
>
> We thank the reviewer for their comments. We are enthused by the reviewer’s excitement about domain privacy.

---

### Meta-Review · Area_Chair_6R5U · 2023-09-23

**Recommendation:** 3

**Metareview:**

The reviewers agree that the notion of domain privacy of general-purpose language models introduced in this paper is a novel and interesting idea.  Also the solid experimental design for evaluating privacy leakage and the results are appreciated.  One important concern emerging from the reviews and discussion is that the authors should take care to clearly formulate the scope of what the work is aiming to do.  The current title suggests too wide a scope to be justified by the work, and therefore the alternative title brought forward by the authors themselves in their rebuttal seems reasonable.

---

### Decision · Program_Chairs · 2023-10-07

**Decision:**

Accept-Findings

**Comment:**

The reviewers agree that the notion of domain privacy of general-purpose language models introduced in this paper is a novel and interesting idea.  Also the solid experimental design for evaluating privacy leakage and the results are appreciated.  One important concern emerging from the reviews and discussion is that the authors should take care to clearly formulate the scope of what the work is aiming to do.  The current title suggests too wide a scope to be justified by the work, and therefore the alternative title brought forward by the authors themselves in their rebuttal seems reasonable.